# EFFICIENT EXPLORATION FOR MODEL-BASED REINFORCEMENT LEARNING WITH CONTINUOUS STATES AND ACTIONS

## ABSTRACT

Balancing exploration and exploitation is crucial in reinforcement learning (RL). In this paper, we study the model-based posterior sampling algorithm in continuous state-action spaces theoretically and empirically. First, we improve the regret bound: with the assumption that reward and transition functions can be modeled as Gaussian Processes with linear kernels, we develop a Bayesian regret bound of $\tilde{O}(H^{3/2}d\sqrt{T})$, where $H$ is the episode length, $d$ is the dimension of the state-action space, and $T$ indicates the total time steps. Our bound can be extended to nonlinear cases as well: using linear kernels on the feature representation $\phi$, the Bayesian regret bound becomes $\tilde{O}(H^{3/2}d_\phi\sqrt{T})$, where $d_\phi$ is the dimension of the representation space. Moreover, we present MPC-PSRL, a model-based posterior sampling algorithm with model predictive control for action selection. To capture the uncertainty in models and realize posterior sampling, we use Bayesian linear regression on the penultimate layer (the feature representation layer $\phi$) of neural networks. Empirical results show that our algorithm achieves the best sample efficiency in benchmark control tasks compared to prior model-based algorithms, and matches the asymptotic performance of model-free algorithms.

## 1 INTRODUCTION

In reinforcement learning (RL), an agent interacts with an unknown environment which is typically modeled as a Markov Decision Process (MDP). Efficient exploration has been one of the main challenges in RL: the agent is expected to balance between *exploring* unseen state-action pairs to gain more knowledge about the environment, and *exploiting* existing knowledge to optimize rewards in the presence of known data.

To achieve efficient exploration, Bayesian reinforcement learning is proposed, where the MDP itself is treated as a random variable with a prior distribution. This prior distribution of the MDP provides an initial uncertainty estimate of the environment, which generally contains distributions of transition dynamics and reward functions. The epistemic uncertainty (subjective uncertainty due to limited data) in reinforcement learning can be captured by posterior distributions given the data collected by the agent.

Posterior sampling reinforcement learning (PSRL), motivated by Thompson sampling in bandit problems (Thompson, 1933), serves as a provably efficient algorithm under Bayesian settings. In PSRL, the agent maintains a posterior distribution for the MDP and follows an optimal policy with respect to a single MDP sampled from the posterior distribution for interaction in each episode. Appealing results of PSRL in tabular RL were presented by both model-based (Osband et al., 2013; Osband & Van Roy, 2017) and model free approaches (Osband et al., 2019) in terms of the Bayesian regret. For $H$-horizon episodic RL, PSRL was proved to achieve a regret bound of $\tilde{O}(H\sqrt{SAT})$, where $S$ and $A$ denote the number of states and actions, respectively. However, in continuous state-action spaces $S$ and $A$ can be infinite, hence the above results do not apply.

Although PSRL in continuous spaces has also been studied in episodic RL, existing results either provide no guarantee or suffer from an exponential order of $H$. In this paper, we achieve the first Bayesian regret bound for posterior sampling algorithms that is near optimal in $T$ (i.e. $\sqrt{T}$) and

polynomial in the episode length $H$ for continuous state-action spaces. We will explain the limitations of previous works in Section 1.1, then summarize our approach and contributions in Section 1.2.

## 1.1 LIMITATIONS OF PREVIOUS BAYESIAN REGRETS IN CONTINUOUS SPACES

**The exponetial order of** $H$: In model-based settings, Osband & Van Roy (2014) derive a regret bound of $\tilde{O}(\sigma_R\sqrt{d_K(R)d_E(R)T} + \mathbb{E}[L^*]\sigma_p\sqrt{d_K(P)d_E(P)})$, where $L^*$ is a global Lipschitz constant for the future value function defined in their eq. (3). However, $L^*$ is dependent on $H$: the difference between input states will propagate in $H$ steps, which results in a term dependent of $H$ in the value function. The authors do not mention this dependency, so there is no clear dependency on $H$ in their regret. Moreover, they use the Lipschitz constant of the underlying value function as an upper bound of $L^*$ in the corollaries, which yields an exponential order in $H$. Take their Corollary 2 of linear quadratic systems as an example: the regret bound is $\tilde{O}(\sigma C\lambda_1 n^2\sqrt{T})$, where $\lambda_1$ is the largest eigenvalue of the matrix $Q$ in the optimal value function $V_1(s) = s^T Q s$. [1] However, the largest eigenvalue of $Q$ is actually exponential in $H$ [2]. Even if we change the reward function from quadratic to linear,the Lipschitz constant of the optimal value function is still exponential in $H$ [3]. Chowdhury & Gopalan (2019) maintains the assumption of this Lipschitz property, thus there exists $\mathbb{E}[L^*]$ with no clear dependency on $H$ in their regret, and in their Corollary 2 of LQR, they follow the same steps as Osband & Van Roy (2014), and still maintain a term with $\lambda_1$, which is actually exponential in $H$ as discussed. Although Osband & Van Roy (2014) mentions that system noise helps to smooth future values, but they do not explore it although the noise is assumed to be subgaussian. The authors directly use the Lipschitz continuity of the underlying function in the analysis of LQR, thus they cannot avoid the exponential term in $H$. Chowdhury & Gopalan (2019) do not explore how the system noise can improve the theoretical bound either. In model-free settings, Azizzadenesheli et al. (2018) develops a regret bound of $\tilde{O}(d_\phi\sqrt{T})$ using a linear function approximator in the Q-network, where $d_\phi$ is the dimension of the feature representation vector of the state-action space, but their bound is still exponential in $H$ as mentioned in their paper.

**High dimensionality**: The eluder dimension of neural networks in Osband & Van Roy (2014) can be infinite, and the information gain (Srinivas et al., 2012) used in Chowdhury & Gopalan (2019) yields exponential order of the state-action spaces dimension $d$ if nonlinear kernels are used, such as SE kernels. However, linear kernels can only model linear functions, thus the representation power is highly restricted if the polynomial order of $d$ is desired.

## 1.2 OUR APPROACH AND MAIN CONTRIBUTIONS

To further imporve the regret bound for PSRL in continuous spaces, especially with explicit dependency on $H$, we study model-based posterior sampling algorithms in episodic RL. We assume that rewards and transitions can be modeled as Gaussian Processes with linear kernels, and extend the assumption to non-linear settings utilizing features extracted by neural networks. For the linear case, we develop a Bayesian regret bound of $\tilde{O}(H^{3/2}d\sqrt{T})$. Using feature embedding technique as mentioned in Yang & Wang (2019), we derive a bound of $\tilde{O}(H^{3/2}d_\phi\sqrt{T})$. Our Bayesian regret is the best-known Bayesian regret for posterior sampling algorithms in continuous state-action spaces, and it also matches the best-known frequentist regret (Zanette et al. (2020), will be discussed in Section 2). Explicitly dependent on $d, H, T$, our result achieves a significant improvement in terms of the Bayesian regret of PSRL algorithms compared to previous works:

**1. We significantly improved the order of** $H$ **to polynomial**: In our analysis, we use the property of subgaussian noise, which is already assumed in Osband & Van Roy (2014) and Chowdhury & Gopalan (2019), to develop a bound with clear polynomial dependency on $H$, without assuming the Lipschitz continuity of the underlying value function. More specifically, we prove Lemma 1, and use

---

[1] $V_1$ denotes the value function counting from step 1 to H within an episode, $s$ is the initial state, reward at the $i$-th step $r_i = s_i^T P s_i + a_i^T R a_i + \epsilon_{P,i}$, and the state at the $i + 1$-th step $s_{i+1} = A s_i + B a_i + \epsilon_{P,i}$ , $i \in [H]$.

[2] Recall the Bellman equation we have $V_i(s_i) = \min_{a_i} s_i^T P s_i + a_i^T R a_i + \epsilon_{P,i} + V_{i+1}(A s_i + B a_i + \epsilon_{P,i})$, $V_{H+1}(s) = 0$ . Thus in $V_1(s)$, there is a term of $(A^{H-1}s)^T P(A^{H-1}s)$, and the eigenvalue of the matrix $(A^{H-1})^T P A^{H-1}$ is exponential in $H$.

[3] For example, if $r_i = s_i^T P + a_i^T R + \epsilon_{P,i}$, there would still exist term of $(A^{H-1}s)^T P$ in $V_1(s)$.

it to develop a clear dependency on $H$, thus we can avoid handling the Lipschitz continuity of the underlying value function.

**2. Lower dimensionality compared to Osband & Van Roy (2014) and Chowdhury & Gopalan (2019)**: We first derive results for linear kernels, and increase the representation power of the linear model by building a Bayesian linear regression model on the feature representation space instead of the original state-action space. As a result, we can use the result of linear kernels to derive a bound linear in the feature dimension. The feature dimension, which in practice is dimension of the last hidden layers in the neural networks required for learning, is much lower than exponential of the input dimension, so we avoid the exponential order of the dimension from the use of nonlinear kernels in Chowdhury & Gopalan (2019).

**3. Fewer assumptions and different proof strategy compared to Chowdhury & Gopalan (2019)**: Although we also use kernelized MDPs like Chowdhury & Gopalan (2019), we omit their assumption A1 (Lipschitz assumption) and A2 (Regularity assumption), only use A3 (subgaussian noise). We avoid A1 since it could be derived from our Lemma 1. Moreover, We directly analyze the regret bound of PSRL using the fact that the sampled and the real unknown MDP share the same distribution conditioned on history. In contrast, Chowdhury & Gopalan (2019) first analyze UCRL (Upper confidence bound in RL) with an extra assumption A2, then transfer it to PSRL.

Empirically, we implement PSRL using Bayesian linear regression (BLR) on the penultimate layer (for feature representation) of neural networks when fitting transition and reward models. We use model predictive control (MPC,Camacho & Alba (2013)) to optimize the policy under the sampled models in each episode as an approximate solution of the sampled MDP as described in Section 5. Experiments show that our algorithm achieves more efficient exploration compared with previous model-based algorithms in control benchmark tasks.

## 2 RELATED WORK ON FREQUENIST REGRETS

Besides the aforementioned works on Bayesian regret bounds, the majority of papers in efficient RL choose the non-Bayesian perspective and develop frequentist regret bounds where the regret for any MDP $M^* \in \mathcal{M}$ is bounded and $M^* \in \mathcal{M}$ holds with high probability. frequentist regret bounds can be expressed in the Bayesian view: for a given confidence set $\mathcal{M}$, the frequentist regret bound implies an identical Bayes regret bound for any prior distribution with support on $\mathcal{M}$. Note that frequentist regret is extensively studied in tabular RL (see Jaksch et al. (2010), Azar et al. (2017), and Jin et al. (2018) as examples), among which the best bound for episodic settings is $\tilde{O}(H\sqrt{SAT})$.

There is also a line of work that develops frequentist bounds with feature representation. Most recently, MatrixRL proposed by (Yang & Wang, 2019) uses low dimensional representation and achieves a regret bound of $\tilde{O}(H^2 d_\phi \sqrt{T})$, which is the best-known frequentist bound in model based settings. While our method is also model-based, we achieve a tighter regret bound when compared in the Bayesian view. In model-free settings, Jin et al. (2020) developed a bound of $\tilde{O}(H^{3/2} d_\phi^{3/2} \sqrt{T})$. Zanette et al. (2020) further improved the regret to $\tilde{O}(H^{3/2} d_\phi \sqrt{T})$ by the proposed an algorithm called ELEANOR, which achieves the best-known frequentist bound in model-free settings. They showed that it is unimprovable with the help of a lower bound established in the bandit literature. Despite that our regret is developed in model-based settings, it matches their bound with the same order of $H$, $d_\phi$ and $T$ in the Bayesian view. Moreover, their algorithm involves optimization over all MDPs in the confidence set, and thus can be computationally prohibitive. Our method is computationally tractable as it is much easier to optimize a single sampled MDP, while matching their regret bound in the Bayesian view.

## 3 PRELIMINARIES

### 3.1 PROBLEM FORMULATION

We model an episodic finite-horizon Markov Decision Process (MDP) $M$ as $\{\mathcal{S}, \mathcal{A}, R^M, P^M, H, \sigma_r, \sigma_f, R_{max}, \rho\}$, where $\mathcal{S} \subset \mathbb{R}^{d_s}$ and $\mathcal{A} \subset \mathbb{R}^{d_a}$ denote state and action spaces, respectively. Each episode with length $H$ has an initial state distribution $\rho$. At time step $i \in [1, H]$ within an episode, the agent observes $s_i \in \mathcal{S}$, selects $a_i \in \mathcal{A}$, receives a noised

reward $r_i \sim R^M(s_i, a_i)$ and transitions to a noised new state $s_{i+1} \sim P^M(s_i, a_i)$. More specifically, $r(s_i, a_i) = \bar{r}^M(s_i, a_i) + \epsilon_r$ and $s_{i+1} = f^M(s_i, a_i) + \epsilon_f$, where $\epsilon_r \sim \mathcal{N}(0, \sigma_r^2)$, $\epsilon_f \sim \mathcal{N}(0, \sigma_f^2 I_{d_s})$. Variances $\sigma_r^2$ and $\sigma_f^2$ are fixed to control the noise level. Without loss of generality, we assume the expected reward an agent receives at a single step is bounded $|\bar{r}^M(s, a)| \leq R_{max}, \forall s \in \mathcal{S}, a \in \mathcal{A}$. Let $\mu: \mathcal{S} \to \mathcal{A}$ be a deterministic policy. Here we define the value function for state $s$ at time step $i$ with policy $\mu$ as $V_{\mu,i}^M(s) = \mathbb{E}[\Sigma_{j=i}^H [\bar{r}^M(s_j, a_j)|s_i = s]$, where $s_{j+1} \sim P^M(s_j, a_j)$ and $a_j = \mu(s_j)$. With the bound expected reward, we have that $|V(s)| \leq HR_{max}, \forall s$.

We use $M^*$ to indicate the real unknown MDP which includes $R^*$ and $P^*$, and $M^*$ itself is treated as a random variable. Thus, we can treat the real noiseless reward function $\bar{r}^*$ and transition function $f^*$ as random processes as well. In the posterior sampling algorithm $\pi^{PS}$, $M^k$ is a random sample from the posterior distribution of the real unknown MDP $M^*$ in the $k$th episode, which includes the posterior samples of $R^k$ and $P^k$, given history prior to the $k$th episode: $\mathcal{H}_k := \{s_{1,1}, a_{1,1}, r_{1,1}, \cdots, s_{k-1,H}, a_{k-1,H}, r_{k-1,H}\}$, where $s_{k,i}, a_{k,i}$ and $r_{k,i}$ indicate the state, action, and reward at time step $i$ in episode $k$. We define the the optimal policy under $M$ as $\mu^M \in_{argmax_\mu} V_{\mu,i}^M(s)$ for all $s \in \mathcal{S}$ and $i \in [H]$. In particular, $\mu^*$ indicates the optimal policy under $M^*$ and $\mu^k$ represents the optimal policy under $M^k$. Let $\Delta_k$ denote the regret over the $k$th episode:

$$\Delta_k = \int \rho(s_1)(V_{\mu^*,1}^{M^*}(s_1) - V_{\mu^k,1}^{M^*}(s_1))ds_1 \tag{1}$$

Then we can express the regret of $\pi^{ps}$ up to time step T as:

$$Regret(T, \pi^{ps}, M^*) := \Sigma_{k=1}^{\lceil \frac{T}{H} \rceil} \Delta_k, \tag{2}$$

Let $BayesRegret(T, \pi^{ps}, \phi)$ denote the Beyesian regret of $\pi^{ps}$ as defined in Osband & Van Roy (2017), where $\phi$ is the prior distribution of $M^*$:

$$BayesRegret(T, \pi^{ps}, \phi) = \mathbb{E}[Regret(T, \pi^{ps}, M^*)]. \tag{3}$$

### 3.2 Assumptions

Generally, we consider modeling an unknown target function $g : \mathbb{R}^d \to \mathbb{R}$. We are given a set of noisy samples $y = [y_1...., y_T]^T$ at points $X = [x_1, ..., x_T]^T$, $X \subset D$, where $D$ is compact and convex, $y_i = g(x_i) + \epsilon_i$ with $\epsilon_i \sim N(0, \sigma^2)$ i.i.d. Gaussian noise $\forall i \in \{1, \cdots, T\}$.

We model $g$ as a sample from a Gaussian Process $GP(\mu(x), \mathcal{K}(x, x'))$, specified by the mean function $\mu(x) = \mathbb{E}[g(x)]$ and the covariance (kernel) function $\mathcal{K}(x, x') = \mathbb{E}[(g(x) - \mu(x))(g(x') - \mu(x'))]$.

Let the prior distribution without any data as $GP(0, \mathcal{K}(x, x'))$. Then the posterior distribution over $g$ given $X$ and $y$ is also a GP with mean $\mu_T(x)$, covariance $\mathcal{K}_T(x, x')$, and variance $\sigma_T^2(x)$: $\mu_T(x) = \mathcal{K}(x, X)(\mathcal{K}(X, X) + \sigma^2 I)^{-1}y$, $\mathcal{K}_T(x, x') = \mathcal{K}(x, x') - \mathcal{K}(X, x)^T(\mathcal{K}(X, X) + \sigma^2 I)^{-1}\mathcal{K}(X, x)$, $\sigma_T^2(x) = \mathcal{K}_T(x, x)$, where $\mathcal{K}(X, x) = [\mathcal{K}(x_1, x), ..., \mathcal{K}(x_T, x)]^T$, $\mathcal{K}(X, X) = [\mathcal{K}(x_i, x_j)]_{1 \leq i \leq T, 1 \leq j \leq T}$.

We model our reward function $\bar{r}^M$ as a Gaussian Process with noise $\sigma_r^2$. For transition models, we treat each dimension independently: each $f_i(s, a), i = 1, .., d_S$ is modeled independently as above, and with the same noise level $\sigma_f^2$ in each dimension. Thus it corresponds to our formulation in the RL setting. Since the posterior covariance matrix is only dependent on the input rather than the target value, the distribution of each $f_i(s, a)$ shares the same covariance matrice and only differs in the mean function.

## 4 Bayesian Regret Analysis

### 4.1 Linear case

**Theorem 1** *In the RL problem formulated in Section 3.1, under the assumption of Section 3.2 with linear kernels[4], we have $BayesRegret(T, \pi^{ps}, M^*) = \tilde{O}(H^{3/2}d\sqrt{T})$, where d is the dimension of the state-action space, H is the episode length, and T is the time elapsed.*

---

[4]GP with linear kernel correspond to Bayesian linear regression $f(x) = w^T x$, where the prior distribution of the weight is $w \sim \mathcal{N}(0, \Sigma_p)$.

*Proof*    The regret in episode $k$ can be rearranged as:

$$\Delta_k = \int \rho(s_1)(V^{M^*}_{\mu^*,}(s_1) - V^{M^k}_{\mu^k,1}(s_1)) + (V^{M^k}_{\mu^k,1}(s_1) - V^{M^*}_{\mu^k,1}(s_1)))ds_1 \tag{4}$$

Note that conditioned upon history $\mathcal{H}_k$ for any $k$, $M^k$ and $M^*$ are identically distributed. Osband & Van Roy (2014) showed that $V^{M^*}_{\mu^*,} - V^{M^k}_{\mu^k,1}$ is zero in expectation, and that only the second part of the regret decomposition need to be bounded when deriving the Bayesian regret of PSRL. Thus we can focus on the policy $\mu_k$, the sampled $M^k$ and real environment data generated by $M^*$. For clarity, the value function $V^{M^k}_{\mu^k,1}$ is simplified to $V^k_{k,1}$ and $V^{M^*}_{\mu^k,1}$ to $V^*_{k,1}$. It suffices to derive bounds for any initial state $s_1$ as the regret bound will still hold through integration of the initial distribution $\rho(s_1)$.

We can rewrite the regret from concentration via the Bellman operator (see Section 5.1 in Osband et al. (2013)):

$$\mathbb{E}[\tilde{\Delta}_k|\mathcal{H}_k] := \mathbb{E}[V^k_{k,1}(s_1) - V^*_{k,1}(s_1)|\mathcal{H}_k]$$

$$= \mathbb{E}[\bar{r}^k(s_1,a_1) - \bar{r}^*(s_1,a_1) + \int P^k(s'|s_1,a_1)V^k_{k,2}(s')ds' - \int P^*(s',|s_1,a_1)V^k_{k,2}(s')ds'|\mathcal{H}_k]$$

$$= \mathbb{E}[\Sigma^H_{i=1}\bar{r}^k(s_i,a_i) - \bar{r}^*(s_i,a_i) + \Sigma^H_{i=1}(\int (P^k(s'|s_i,a_i) - P^*(s'|s_i,a_i))V^k_{k,i+1}(s')ds')|\mathcal{H}_k]$$

$$= \mathbb{E}[\tilde{\Delta}_k(r) + \tilde{\Delta}_k(f)|\mathcal{H}_k]$$

$$\tag{5}$$

where $a_i = \mu_k(s_i), s_{i+1} \sim P^*(s_{i+1}|s_i,a_i), \tilde{\Delta}_k(r) = \Sigma^H_{i=1}\bar{r}^k(s_i,a_i) - \bar{r}^*(s_i,a_i), \tilde{\Delta}_k(f) = \Sigma^H_{i=1}(\int (P^k(s'|s_i,a_i) - P^*(s'|s_i,a_i))V^k_{k,i+1}(s')ds')$. Thus, here $(s_i,a_i)$ is the state-action pair that the agent encounters in the $k$th episode while using $\mu_k$ for interaction in the real MDP $M^*$. We can define $V_{k,H+1} = 0$ to keep consistency. Note that we cannot treat $s_i$ and $a_i$ as deterministic and only take the expectation directly on random reward and transition functions. Instead, we need to bound the difference using concentration properties of reward and transition functions modeled as Gaussian Processes (which also applies to any state-action pair), and then derive bounds of this expectation. For all $i$, we have $\int (P^k(s'|s_i,a_i) - P^*(s'|s_i,a_i))V^k_{k,i+1}(s')ds' \le \max_s |V^k_{k,i+1}(s)| \int |P^k(s'|s_i,a_i) - P^*(s'|s_i,a_i)|ds' \le HR_{max} \int |P^k(s'|s_i,a_i) - P^*(s'|s_i,a_i)|ds'$.

Now we present a lemma which enables us to derive a regret bound with explicit dependency on the episode length $H$.

**Lemma 1** *For two multivariate Gaussian distribution $\mathcal{N}(\boldsymbol{\mu}, \sigma^2 I)$, $\mathcal{N}(\boldsymbol{\mu'}, \sigma^2 I)$ with probability density function $p_1(\boldsymbol{x})$ and $p_2(\boldsymbol{x})$ respectively, $\boldsymbol{x} \in \mathbb{R}^d$,*

$$\int |p_1(\boldsymbol{x}) - p_2(\boldsymbol{x})|d\boldsymbol{x} \le \sqrt{\frac{2}{\pi\sigma^2}}||\boldsymbol{\mu} - \boldsymbol{\mu'}||_2.$$

The proof is in Appendix A.1. Clearly, this result can also be extended to sub-Gaussian noises.

Recall that $P^k(s'|s_i,a_i) = \mathcal{N}(f^k(s_i,a_i), \sigma_f^2 I)$ and $P^*(s'|s_i,a_i) = \mathcal{N}(f^*(s_i,a_i), \sigma_f^2 I)$. By Lemma 1 we have

$$\int |P^k(s'|s_i,a_i) - P^*(s'|s_i,a_i)|ds' \le \sqrt{\frac{2}{\pi\sigma_f^2}}||f^k(s_i,a_i) - f^*(s_i,a_i)||_2 \tag{6}$$

**Lemma 2** *(Rigollet & Hütter, 2015) Let $X_1, ..., X_N$ be $N$ sub-Gaussian random variables with variance $\sigma^2$ (not required to be independent). Then for any $t > 0$, $\mathbb{P}(\max_{1 \le i \le N} |X_i| > t) \le 2Ne^{-\frac{t^2}{2\sigma^2}}$.*

Given history $\mathcal{H}_k$, let $\bar{f}^k(s,a)$ indicate the posterior mean of $f^k(s,a)$ in episode $k$, and $\sigma_k^2(s,a)$ denotes the posterior variance of $f^k$ in each dimension. Note that $f^*$ and $f^k$ share the same variance in each dimension given history $\mathcal{H}_k$, as described in Section 3. Consider all dimensions of the state space, by Lemma 2, we have that with probability at least $1 - \delta$, $\max_{1 \le i \le d_s} |f^k_i(s,a) - $

$\bar{f}_i^k(s,a)| \leq \sqrt{2\sigma_k^2(s,a)log\frac{2d_s}{\delta}}$. Also, we can derive an upper bound for the norm of the state difference $||f^k(s,a) - \bar{f}^k(s,a)||_2 \leq \sqrt{d_s}\max_{1\leq i\leq d_s}|f_i^k(s,a) - \bar{f}_i^k(s,a)|$, and so does $||f^*(s,a) - \bar{f}^k(s,a)||_2$ since $f^*$ and $f^k$ share the same posterior distribution. By the union bound, we have that with probability at least $1 - 2\delta$ $||f^k(s,a) - f^*(s,a)||_2 \leq 2\sqrt{2d_s\sigma_k^2(s,a)log\frac{2d_s}{\delta}}$.

Then we look at the sum of the differences over horizon $H$, without requiring each variable in the sum to be independent:

$$\mathbb{P}(\Sigma_{i=1}^H||f^k(s_i,a_i) - f^*(s_i,a_i)||_2 > \Sigma_{i=1}^H 2\sqrt{2d_s\sigma_k^2(s_i,a_i)log\frac{2d_s}{\delta}})$$

$$\leq \mathbb{P}(\bigcup_{i=1}^H\{||f^k(s_i,a_i) - f^*(s_i,a_i)||_2 > 2\sqrt{2d_s\sigma_k^2(s_i,a_i)log\frac{2d_s}{\delta}}\}) \qquad (7)$$

$$\leq \Sigma_{i=1}^H\mathbb{P}(||f^k(s_i,a_i) - f^*(s_i,a_i)||_2 > 2\sqrt{2d_s\sigma_k^2(s_i,a_i)log\frac{2d_s}{\delta}})$$

Thus, with probability at least $1 - 2H\delta$, we have $\Sigma_{i=1}^H||f^k(s_i,a_i) - f^*(s_i,a_i)||_2 \leq \Sigma_{i=1}^H 2\sqrt{2d_s\sigma_k^2(s_i,a_i)log\frac{2d_s}{\delta}}$. Let $\delta' = 2H\delta$, we have that with probability $1 - \delta$, $\Sigma_{i=1}^H||f^k(s_i,a_i) - f^*(s_i,a_i)||_2 \leq \Sigma_{i=1}^H 2\sqrt{2d_s\sigma_k^2(s_i,a_i)log\frac{4Hd_s}{\delta}} \leq 2H\sqrt{2d_s\sigma_k^2(s_{k_{max}},a_{k_{max}})log\frac{4Hd_s}{\delta}}$, where the index $k_{max} = \arg\max_i \sigma_k(s_i,a_i), i = 1,...,H$ in episode $k$. Here, since the posterior distribution is only updated every H steps, we have to use data points with the max variance in each episode to bound the result. Similarly, using the union bound for $[\frac{T}{H}]$ episodes, and let $C = \sqrt{\frac{2}{\pi\sigma_f^2}}$, we have that with probability at least $1 - \delta$, $\Sigma_{k=1}^{[\frac{T}{H}]}[\tilde{\Delta}_k(f)|\mathcal{H}_k] \leq \Sigma_{k=1}^{[\frac{T}{H}]}\Sigma_{i=1}^H 2CHR_{max}||f^k(s_i,a_i) - f^*(s_i,a_i)||_2 \leq \Sigma_{k=1}^{[\frac{T}{H}]}4CH^2R_{max}\sqrt{2d_s\sigma_k^2(s_{k_{max}},a_{k_{max}})log\frac{4Td_s}{\delta}}$.

In each episode $k$, let $\sigma_k'^2(s,a)$ denote the posterior variance given only a subset of data points $\{(s_{1_{max}},a_{1_{max}}),...,(s_{k-1_{max}},a_{k-1_{max}})\}$, where each element has the max variance in the corresponding episode. By Eq.(6) in Williams & Vivarelli (2000), we know that the posterior variance reduces as the number of data points grows. Hence $\forall(s,a), \sigma_k^2(s,a) \leq \sigma_k'^2(s,a)$. By Theorem 5 in Srinivas et al. (2012) which provides a bound on the information gain, and Lemma 2 in Russo & Van Roy (2014) that bounds the sum of variances by the information gain, we have that $\Sigma_{k=1}^{[\frac{T}{H}]}\sigma_k'^2(s_{k_{max}},a_{k_{max}}) = \mathcal{O}((d_s + d_a)log[\frac{T}{H}])$ for linear kernels with bounded variances. Note that the bounded variance property for linear kernels only requires the range of all state-action pairs actually encountered in $M^*$ not to expand to infinity as T grows, which holds in general episodic MDPs.

Thus with probability $1 - \delta$, and let $\delta = \frac{1}{T}$,

$$\Sigma_{k=1}^{[\frac{T}{H}]}[\tilde{\Delta}_k(f)|\mathcal{H}_k] \leq \Sigma_{k=1}^{[\frac{T}{H}]}4CH^2R_{max}\sqrt{2d_s\sigma_k^2(s_{k_{max}},a_{k_{max}})log\frac{4Td_s}{\delta}}$$

$$\leq \Sigma_{k=1}^{[\frac{T}{H}]}8CH^2R_{max}\sqrt{d_s\sigma_k'^2(s_{k_{max}},a_{k_{max}})log(2Td_s)}$$

$$\leq 8CH^2R_{max}\sqrt{\Sigma_{k=1}^{[\frac{T}{H}]}\sigma_k'^2(s_{k_{max}},a_{k_{max}})}\sqrt{[\frac{T}{H}]}\sqrt{d_slog(2Td_s)} \qquad (8)$$

$$= 8CH^{\frac{3}{2}}R_{max}\sqrt{T}\sqrt{d_slog(2Td_s)} * \sqrt{\mathcal{O}((d_s + d_a)log[\frac{T}{H}])} = \tilde{\mathcal{O}}((d_s + d_a)H^{\frac{3}{2}}\sqrt{T})$$

where $\tilde{\mathcal{O}}$ ignores logarithmic factors.

Therefore, $\mathbb{E}[\Sigma_{k=1}^{[\frac{T}{H}]}\tilde{\Delta}_k(f)|\mathcal{H}_k] \leq (1 - \frac{1}{T})\tilde{\mathcal{O}}((d_s + s_a)H^{\frac{3}{2}}T) + \frac{1}{T}2HR_{max}*[\frac{T}{H}] = \tilde{\mathcal{O}}(H^{\frac{3}{2}}d\sqrt{T})$, where $2HR_{max}$ is the upper bound on the difference of value functions, and $d = d_s + d_a$. By similar derivation, $\mathbb{E}[\Sigma_{k=1}^{[\frac{T}{H}]}\tilde{\Delta}_k(r)|\mathcal{H}_k] = \tilde{\mathcal{O}}(\sqrt{dHT})$. Finally, through the tower property we have $BayesRegret(T,\pi^{ps},M^*) = \tilde{\mathcal{O}}(H^{\frac{3}{2}}d\sqrt{T})$.

$\square$

---

**Algorithm 1** MPC-PSRL

---

Initialize data $\mathcal{D}$ with random actions for one episode
**repeat**
    Sample a transition model and a cost model at the beginning of each episode
    **for** $i = 1$ to $H$ steps **do**
        Obtain action using MPC with planning horizon $\tau$: $a_i \in \arg\max_{a_{i:i+\tau}} \sum_{t=i}^{i+\tau} \mathbb{E}[r(s_t, a_t)]$
        $\mathcal{D} = \mathcal{D} \cup \{(s_i, a_i, r_i, s_{i+1})\}$
    **end for**
    Train cost and dynamics representations $\phi_r$ and $\phi_f$ using data in $\mathcal{D}$
    Update $\phi_r(s, a)$, $\phi_f(s, a)$ for all $(s, a)$ collected
    Perform posterior update of $w_r$ and $w_f$ in cost and dynamics models using updated representations $\phi_r(s, a)$, $\phi_f(s, a)$ for all $(s, a)$ collected
**until** convergence

---

## 4.2 Nonlinear case via feature representation

We can slightly modify the previous proof to derive the bound in settings that use feature representations. We can transform the state-action pair $(s, a)$ to $\phi_f(s, a) \in \mathbb{R}^{d_\phi}$ as the input of the transition model , and transform the newly transitioned state $s'$ to $\psi_f(s') \in \mathbb{R}^{d_\psi}$ as the target, then the transition model can be established with respect to this feature embedding. We further assume $d_\psi = O(d_\phi)$ as Assumption 1 in Yang & Wang (2019). Besides, we assume $d_{\phi'} = O(d_\phi)$ in the feature representation $\phi_r(s, a) \in \mathbb{R}^{d_{\phi'}}$, then the reward model can also be established with respect to the feature embedding. Following similar steps, we can derive a Bayesian regret of $\tilde{O}(H^{3/2} d_\phi \sqrt{T})$.

## 5 Algorithm Description

In this section, we elaborate our proposed algorithm, MPC-PSRL, as shown in Algorithm 1.

### 5.1 Predictive model

When model the rewards and transitions, we use features extracted from the penultimate layer of fitted neural networks, and perform Bayesian linear regression on the feature vectors to update posterior distributions.

**Feature representation:** we first fit neural networks for transitions and rewards, using the same network architecture as Chua et al. (2018). Let $x_i$ denote the state-action pair $(s_i, a_i)$ and $y_i$ denote the target value. Specifically, we use reward $r_i$ as $y_i$ to fit rewards, and we take the difference between two consecutive states $s_{i+1} - s_i$ as $y_i$ to fit transitions. The penultimate layer of fitted neural networks is extracted as the feature representation, denoted as $\phi_f$ and $\phi_r$ for transitions and rewards, respectively. Note that in the transition feature embedding, we only use one neural network to extract features of state-action pairs from the penultimate layer to serve as $\phi$, and leave the target states without further feature representation (the general setting is discussed in Section 4.2 where feature representations are used for both inputs and outputs), so the dimension of the target in the transition model $d(\psi)$ equals to $d_s$. Thus we have a modified regret bound of $\tilde{O}(H^{3/2} \sqrt{d d_\phi T})$. We do not find the necessity to further extract feature representations in the target space, as it might introduce additional computational overhead. Although higher dimensionality of the hidden layers might imply better representation, we find that only modifying the width of the penultimate layer to $d_\phi = d_s + s_a$ suffices in our experiments for both reward and transition models. Note that how to optimize the dimension of the penultimate layer for more efficient feature representation deserves further exploration.

**Bayesian update and posterior sampling:** here we describe the Bayesian update of transition and reward models using extracted features. Recall that Gaussian process with linear kernels is equivalent to Bayesian linear regression. By extracting the penultimate layer as feature representation $\phi$, the target value $y$ and the representation $\phi(x)$ could be seen as linearly related: $y = w^\top \phi(x) + \epsilon$, where $\epsilon$ is a zero-mean Gaussian noise with variance $\sigma^2$ (which is $\sigma_f^2$ for the transition model and $\sigma_r^2$ for the reward model as defined in Section 3.1). We choose the prior distribution of weights $w$ as zero-mean

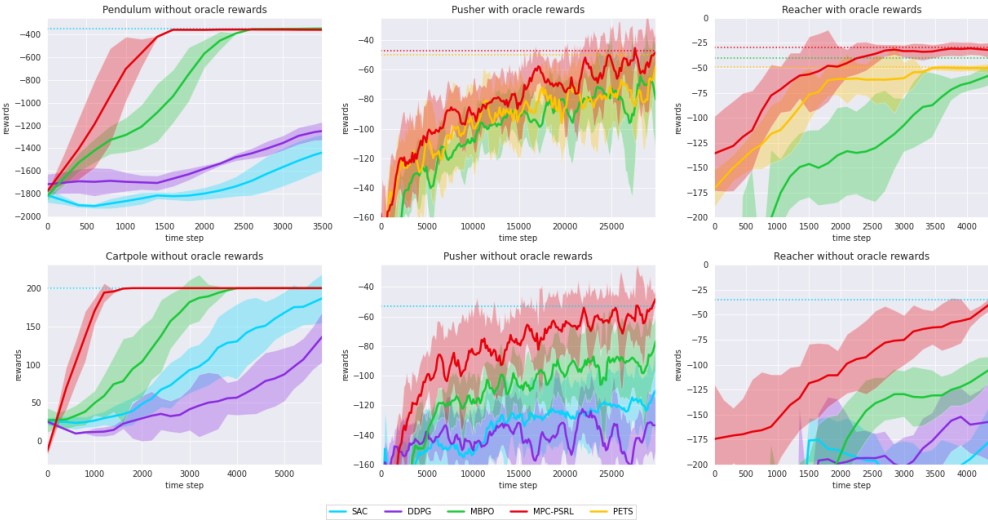

Figure 1: Training curves of MPC-PSRL (shown in red), and other baseline algorithms in different tasks. Solid curves are the mean of five trials, shaded areas correspond to the standard deviation among trials, and the doted line shows the rewards at convergence.

Gaussian with covariance matrix $\Sigma_p$, then the posterior distribution of $w$ is also multivariate Gaussian (Rasmussen (2003)):

$$p(w|\mathcal{D}) \sim \mathcal{N}\left(\sigma^{-2}A^{-1}\Phi Y, A^{-1}\right)$$

where $A = \sigma^{-2}\Phi\Phi^\top + \Sigma_p^{-1}$, $\Phi \in \mathcal{R}^{d \times N}$ is the concatenation of feature representations $\{\phi(x_i)\}_{i=1}^N$, and $Y \in \mathcal{R}^N$ is the concatenation of target values. At the beginning of each episode, we sample $w$ from the posterior distribution to build the model, collect new data during the whole episode, and update the posterior distribution of $w$ at the end of the episode using all the data collected.

Besides the posterior distribution of $w$, the feature representation $\phi$ is also updated in each episode with new data collected. We adopt a similar dual-update procedure as Riquelme et al. (2018): after representations for rewards and transitions are updated, feature vectors of all state-action pairs collected are re-computed. Then we apply Bayesian update on these feature vectors. See the description of Algorithm 1 for details.

## 5.2 PLANNING

During interaction with the environment, we use a MPC controller (Camacho & Alba (2013)) for planning. At each time step $i$, the controller takes state $s_i$ and an action sequence $a_{i:i+\tau} = \{a_i, a_{i+1}, \cdots, a_{i+\tau}\}$ as the input, where $\tau$ is the planning horizon. We use transition and reward models to produce the first action $a_i$ of the sequence of optimized actions $\arg\max_{a_{i:i+\tau}} \sum_{t=i}^{i+\tau} \mathbb{E}[r(s_t, a_t)]$, where the expected return of a series of actions can be approximated using the mean return of several particles propagated with noises of our sampled reward and transition models. To compute the optimal action sequence, we use CEM (Botev et al. (2013)), which samples actions from a distribution closer to previous action samples with high rewards.

## 6 EXPERIMENTS

We compare our method with the following state-of-the art model-based and model-free algorithms on benchmark control tasks.

**Model-free:** Soft Actor Critic (SAC) from Haarnoja et al. (2018) is an off-policy deep actor-critic algorithm that utilizes entropy maximization to guide exploration. Deep Deterministic Policy Gradient (DDPG) from Barth-Maron et al. (2018) is an off-policy algorithm that concurrently learns a Q-function and a policy, with a discount factor to guide exploration.

**Model-based:** Probabilistic Ensembles with Trajectory Sampling (PETS) from Chua et al. (2018) models the dynamics via an ensemble of probabilistic neural networks to capture epistemic uncertainty for exploration, and uses MPC for action selection, with a requirement to have access to oracle rewards for planning. Model-Based Policy Optimization (MBPO) from Janner et al. (2019) uses the same bootstrap ensemble techniques as PETS in modeling, but differs from PETS in policy optimization with a large amount of short model-generated rollouts, and can cope with environments with no oracle rewards provided. We do not compare with Gal et al. (2016), which adopts a single Bayesian neural network (BNN) with moment matching, as it is outperformed by PETS that uses an ensemble of BNNs with trajectory sampling. And we don't compare with GP-based trajectory optimization methods with real rewards provided (Deisenroth & Rasmussen, 2011; Kamthe & Deisenroth, 2018), which are not only outperformed by PETS, but also computationally expensive and thus are limited to very small state-action spaces.

We use environments with various complexity and dimensionality for evaluation. Low-dimensional environments: continuous Cartpole ($d_s = 4$, $d_a = 1, H = 200$, with a continuous action space compared to the classic Cartpole, which makes it harder to learn) and Pendulum Swing Up ($d_s = 3$, $d_a = 1, H = 200$, a modified version of Pendulum where we limit the start state to make it harder for exploration). Trajectory optimization with oracle rewards in these two environments is easy and there is almost no difference in the performances for all model-based algorithms we compare, so we omit showing these learning curves. Higher dimensional environments: 7-DOF Reacher ($d_s = 17, d_a = 7, H = 150$) and 7-DOF pusher ($d_s = 20, d_a = 7, H = 150$) are two more challenging tasks as provided in Chua et al. (2018), where we conduct experiments both with and without true rewards, to compare with all baseline algorithms mentioned.

The learning curves of these algorithms are showed in Figure 1. When the oracle rewards are provided in Pusher and Reacher, our method outperforms PETS and MBPO: it converges more quickly with similar performance at convergence in Pusher, while in Reacher, not only does it learn faster but also performs better at convergence. As we use the same planning method (MPC) as PETS, results indicate that our model better captures uncertainty, which is beneficial to improving sample efficiency. When exploring in environments where both rewards and transition are unknown, our method learns significantly faster than previous model-based and model-free methods which do no require oracle rewards. Meanwhile, it matches the performance of SAC at convergence. Moreover, the performances of our algorithm in environments with and without oracle rewards can be similar, or even faster convergence (see Pusher with and without rewards), indicating that our algorithm excels at exploring both rewards and transitions.

From experimental results, it can be verified that our algorithm better captures the model uncertainty, and makes better use of uncertainty using posterior sampling. In our methods, by sampling from a Bayesian linear regression on a fitted feature space, and optimizing under the same sampled MDP in the whole episode instead of re-sampling at every step, the performance of our algorithm is guaranteed from a Bayesian view as analysed in Section 4. While PETS and MBPO use bootstrap ensembles of models with a limited ensemble size to "simulate" a Bayesian model, in which the convergence of the uncertainty is not guaranteed and is highly dependent on the training of the neural network. However, in our method there is a limitation of using MPC, which might fail in even higher-dimensional tasks shown in Janner et al. (2019). Incorporating policy gradient techniques for action-selection might further improve the performance and we leave it for future work.

## 7  CONCLUSION

In our paper, we derive a novel Bayesian regret for PSRL algorithm in continuous spaces with the assumption that true rewards and transitions (with or without feature embedding) can be modeled by GP with linear kernels. While matching the best-known bounds in previous works from a Bayesian view, PSRL also enjoys computational tractability. Moreover, we propose MPC-PSRL in continuous environments, and experiments show that our algorithm exceeds existing model-based and model-free methods with more efficient exploration.

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

## A  APPENDIX

### A.1  PROOF OF LEMMA 1

Here we provide a proof of Lemma 1.

We first prove the results in $\mathbb{R}^d$ with $d = 1$: $p_1(x) \sim \mathcal{N}(\mu, \sigma^2), p_2(x) \sim \mathcal{N}(\mu', \sigma^2)$, without loss of generality, assume $\mu' \geq \mu$. The probabilistic distribution is symmetric with regard to $\frac{\mu+\mu'}{2}$. Note that $p_1(x) = p_2(x)$ at $x = \frac{\mu+\mu'}{2}$. Thus the integration of absolute difference between pdf of $p_1$ and $p_2$ can be simplified as twice the integration of one side:

$$\int_{-\infty}^{\infty} |p_2(x) - p_1(x)| dx = \frac{2}{\sqrt{2\pi\sigma^2}} \int_{\frac{\mu+\mu'}{2}}^{\infty} e^{\frac{-(x-\mu')^2}{2\sigma^2}} - e^{\frac{-(x-\mu)^2}{2\sigma^2}} dx \tag{9}$$

let $z_1 = x - \mu, z_2 = x - \mu'$, we have:

$$\begin{aligned}
&\frac{2}{\sqrt{2\pi\sigma^2}} \int_{\frac{\mu+\mu'}{2}}^{\infty} e^{\frac{-(x-\mu')^2}{2\sigma^2}} - e^{\frac{-(x-\mu)^2}{2\sigma^2}} dx \\
&= \sqrt{\frac{2}{\pi\sigma^2}} \int_{\frac{\mu-\mu'}{2}}^{\infty} e^{-\frac{z_2^2}{2\sigma^2}} dz_2 - \sqrt{\frac{2}{\pi\sigma^2}} \int_{\frac{\mu'-\mu}{2}}^{\infty} e^{-\frac{z_1^2}{2\sigma^2}} dz_1 \\
&= \sqrt{\frac{2}{\pi\sigma^2}} \int_{\frac{\mu-\mu'}{2}}^{\frac{\mu'-\mu}{2}} e^{-\frac{z^2}{2\sigma^2}} dz \\
&= 2\sqrt{\frac{2}{\pi\sigma^2}} \int_{0}^{\frac{\mu'-\mu}{2}} e^{-\frac{z^2}{2\sigma^2}} dz \leq 2\sqrt{\frac{2}{\pi\sigma^2}} \int_{0}^{\frac{\mu'-\mu}{2}} 1 dz = \sqrt{\frac{2}{\pi\sigma^2}} |\mu' - \mu|.
\end{aligned} \tag{10}$$

Now we extend the result to $\mathbb{R}^d(d \geq 2)$: $p_1(x) \sim \mathcal{N}(\mu, \sigma^2 I), p_2(x) \sim \mathcal{N}(\mu', \sigma^2 I)$. We can rotate the coordinate system recursively to align the last axis with vector $\mu - \mu'$, such that the coordinates of $\mu$ and $\mu'$ can be written as $(0, 0, \cdots, 0, \hat{\mu})$, and $(0, 0, \cdots, 0, \hat{\mu}')$ respectively, with $|\hat{\mu}' - \hat{\mu}| = \|\mu - \mu'\|_2$. Without loss of generality, let $\hat{\mu} \geq \hat{\mu}'$.

Clearly, all points with equal distance to $\hat{\mu}'$ and $\hat{\mu}$ define a hyperplane $P : x_d = \frac{\hat{\mu}+\hat{\mu}'}{2}$ where $p_1(x) = p_2(x), \forall x \in P$. More specifically, the probabilistic distribution is symmetric with regard to $P$. Similar to the analysis in $\mathbb{R}^1$:

$$
\begin{aligned}
&\int_{-\infty}^{\infty} \int_{-\infty}^{\infty} \cdots \int_{-\infty}^{\infty} |p_1(x) - p_2(x)| dx_1 dx_2 \cdots dx_d \\
&= \frac{2}{\sqrt{(2\pi)^d \sigma^{2d}}} \int_{-\infty}^{\infty} \int_{-\infty}^{\infty} \cdots \int_{\frac{\hat{\mu}+\hat{\mu}'}{2}}^{\infty} e^{\frac{-x_1^2}{2\sigma^2}} \cdots e^{\frac{-x_{d-1}^2}{2\sigma^2}} e^{\frac{-(x_d-\hat{\mu})^2}{2\sigma^2}} dx_1 dx_2 \cdots dx_d \\
&\quad - \frac{2}{\sqrt{(2\pi)^d \sigma^{2d}}} \int_{-\infty}^{\infty} \int_{-\infty}^{\infty} \cdots \int_{\frac{\hat{\mu}+\hat{\mu}'}{2}}^{\infty} e^{\frac{-x_1^2}{2\sigma^2}} \cdots e^{\frac{-x_{d-1}^2}{2\sigma^2}} e^{\frac{-(x_d-\hat{\mu}')^2}{2\sigma^2}} dx_1 dx_2 \cdots dx_d \\
&= \frac{2}{\sqrt{(2\pi)^d \sigma^{2d}}} \int_{-\infty}^{\infty} e^{\frac{-x_1^2}{2\sigma^2}} dx_1 \int_{-\infty}^{\infty} e^{\frac{-x_2^2}{2\sigma^2}} dx_2 \cdots \int_{-\infty}^{\infty} e^{\frac{-x_{d-1}^2}{2\sigma^2}} dx_{d-1} \int_{\frac{\hat{\mu}+\hat{\mu}'}{2}}^{\infty} e^{\frac{-(x_d-\hat{\mu})^2}{2\sigma^2}} dx_d \\
&\quad - \frac{2}{\sqrt{(2\pi)^d \sigma^{2d}}} \int_{-\infty}^{\infty} e^{\frac{-x_1^2}{2\sigma^2}} dx_1 \int_{-\infty}^{\infty} e^{\frac{-x_2^2}{2\sigma^2}} dx_2 \cdots \int_{-\infty}^{\infty} e^{\frac{-x_{d-1}^2}{2\sigma^2}} dx_{d-1} \int_{\frac{\hat{\mu}+\hat{\mu}'}{2}}^{\infty} e^{\frac{-(x_d-\hat{\mu}')^2}{2\sigma^2}} dx_d \\
&= \sqrt{\frac{2}{\pi\sigma^2}} \left( \int_{\frac{\hat{\mu}+\hat{\mu}'}{2}}^{\infty} e^{\frac{-(x_d-\hat{\mu})^2}{2\sigma^2}} dx_d - \int_{\frac{\hat{\mu}+\hat{\mu}'}{2}}^{\infty} e^{\frac{-(x_d-\hat{\mu}')^2}{2\sigma^2}} dx_d \right)
\end{aligned}
\tag{11}
$$

let $z_1 = x_d - \hat{\mu}, z_2 = x_d - \hat{\mu}'$, we have:

$$
\begin{aligned}
&\int_{\frac{\hat{\mu}+\hat{\mu}'}{2}}^{\infty} e^{\frac{-(x_d-\hat{\mu})^2}{2\sigma^2}} dx_d - \int_{\frac{\hat{\mu}+\hat{\mu}'}{2}}^{\infty} e^{\frac{-(x_d-\hat{\mu}')^2}{2\sigma^2}} dx_d \\
&= \int_{\frac{\hat{\mu}'-\hat{\mu}}{2}}^{\infty} e^{\frac{-z_1^2}{2\sigma^2}} dz_1 - \int_{\frac{\hat{\mu}-\hat{\mu}'}{2}}^{\infty} e^{\frac{-z_2^2}{2\sigma^2}} dz_2 \\
&= \int_{\frac{\hat{\mu}'-\hat{\mu}}{2}}^{\frac{\hat{\mu}-\hat{\mu}'}{2}} e^{\frac{-z_2^2}{2\sigma^2}} dz \\
&= 2 \int_{0}^{\frac{\hat{\mu}-\hat{\mu}'}{2}} e^{\frac{-z_2^2}{2\sigma}} dz \leq 2 \int_{0}^{\frac{\hat{\mu}-\hat{\mu}'}{2}} 1 dz \\
&= |\hat{\mu} - \hat{\mu}'|
\end{aligned}
\tag{12}
$$

Thus $\int_{-\infty}^{\infty} \int_{-\infty}^{\infty} \cdots \int_{-\infty}^{\infty} |p_1(x) - p_2(x)| dx_1 dx_2 \cdots dx_d \leq \sqrt{\frac{2}{\pi\sigma^2}} \|\mu - \mu'\|_2$.

## A.2 EXPERIMENTAL DETAILS

Here we provide hyperparameters for MBPO:

| env | cartpole | pendulum | pusher | reacher |
|---|---|---|---|---|
| env steps per episode | 200 | 200 | 150 | 150 |
| model rollouts per env step | 400 | | | |
| ensemble size | 5 | | | |
| network architecture | MLP with 2 hidden layers of size 200 | MLP with 2 hidden layers of size 200 | MLP with 4 hidden layers of size 200 | MLP with 4 hidden layers of size 200 |
| policy updates per env step | 40 | | | |
| model horizon | 1->15 from episode 1->30 | 1->15 from episode 1->30 | 1 | 1->15 from episode 1->30 |

Table 1: Hyperparamters for MBPO

And we provide hyperparamters for MPC and Neural Networks in PETS:

| env | pusher | reacher |
|---|---|---|
| env steps per episode | 150 | 150 |
| popsize | 500 | 400 |
| number of elites | 50 | 50 |
| network architecture | MLP with 4 hidden layers of size 200 | |
| planning horizon | 30 | 30 |
| max iter | 5 | |
| ensemble size | 5 | |

Table 2: Hyperparamters for PETS

Here are hyperparameters of our algorithm, which is similar with PETS, except for ensemble size(since we do not use ensembled models):

| env | cartpole | pendulum | pusher | reacher |
|---|---|---|---|---|
| env steps per episode | 200 | 200 | 150 | 150 |
| popsize | 500 | 100 | 500 | 400 |
| number of elites | 50 | 5 | 50 | 50 |
| network architecture | MLP with 2 hidden layers of size 200 | MLP with 2 hidden layers of size 200 | MLP with 4 hidden layers of size 200 | MLP with 4 hidden layers of size 200 |
| planning horizon | 30 | 20 | 30 | 30 |
| max iter | 5 | | | |

Table 3: Hyperparamters for our method

For SAC and DDPG, we use the open source code (`https://github.com/dongminlee94/deep_rl`) for implementation without changing their hyperparameters. We appreciate the authors for sharing the code!

