# OpenReview forum: "Efficient Exploration for Model-based Reinforcement Learning with Continuous States and Actions"
_ICLR.cc/2021/Conference — Reject_

### Official Review · AnonReviewer2 · 2020-10-28
**Good approach but not clear contribution in model-based RL**

**Rating:** 6
**Confidence:** 4

**Review:**

Review
This paper proposes a new model-based reinforcement learning algorithm named MPC-PSRL. Theoretically, the authors provide regret analysis of the proposed algorithm. The authors also provide empirical results that MPC-PSRL outperforms other previous model-based RL algorithms, such as PETS or MBPO.
However, it is not clear that the main contribution of this paper. Osband & Van Roy (2014) already provides the algorithm posterior sampling RL algorithm, named PSRL for continuous domain. If posterior distribution of MDP is modeled as GP and optimal policy is computed by MPC, then it is the same as the method in this paper. They also provide regret analysis for continuous domain with sub-Gaussian noise model. What are the major difficulties / differences compared to the previous work?
Questions:
Is there any reason there are no empirical results on Pendulum and Cartpole with oracle rewards?
To emphasize the effect of posterior sampling (maybe exploration), would you provide the results that using just mean instead of sampling? It is not clear whether both GP modeling and exploration via posterior sampling have a significant impact on performance.
Typos:
Use \sigma_R with \sigma_r / \sigma_P with \sigma_f
Theorem 1 in page 3, d_\phi should be d.
In proof of theorem 1, what is the definition of \delta_k(r) and \delta_k(f)?

score: 5 -> 6

---

> ### Author Response · Authors · 2020-11-18
> **Theoretical contribution, comparison with previous works, and performance comparison between mean and sampling**
>
> Thanks for your review and detailed comments! We respectfully disagree that our theoretical novelty is unclear.
>
> **Re 1: Difference between regret in Osband & Van Roy (2014) and ours**
> The main limitation of previous methods (Osband & Van Roy (2014), Chowdhury&Gopalan(2019)) is that their bounds can be exponential in $H$. Please check our theoretical novelty compared to previous methods in the general comments above for details: **https://openreview.net/forum?id=asLT0W1w7Li&noteId=rJqD4DMG8jK**, where we show **why the regret bound in Osband & Van Roy (2014) and Chowdhury&Gopalan(2019) can be exponential in $H$ in continuous spaces in part 1.1, and clarify our improvement with respect to $H$ in part 1.2**. We also illustrate our improvement w.r.t. the dimension $d$ in part 2 and highlight other differences between our proof and Chowdhury&Gopalan(2019) in part 3. Our Bayesian regret is $\tilde{O}(H^{3/2}d\sqrt{T})$, which is the best-known Bayesian regret for posterior sampling algorithms in continuous state-action spaces, and it also matches the best-known frequentist regret (Zanette et al.(2020) in our Related Work Section 2).
>
> **Re 2: Why there is no result on Cartpole and Pendulum with oracle rewards**
> As we mentioned in the fourth paragraph in Section 6 Experiments, trajectory optimization with oracle rewards in Cartpole and Pendulum is **very easy**, and there is almost no difference in the performance for all model-based algorithms we compare, so we omit showing these trivial learning curves.
>
> **Re 3: Performance of sampling verse mean**
> Thanks for your suggestion! We have added a comparison between posterior sampling and using the mean of the same Bayesian linear regression model, which shows that sampling further improves the performance than simply using the mean value. The results are shown below.
>
> |          | cartpole  | pendulum  | pusher       | reacher    | pusher(r)     | reacher(r) |
> |----------|-----------|-----------|--------------|------------|--------------|------------|
> | mean     | 8.6±0.75 | 9.6±0.49 | 103.4±14.59 | 34.7±2.31 | 118.2±7.29  | 11.6±1.36 |
> | sampling | 6.8±1.36 | 5.0±1.41 | 95.8±25.29  | 23.2±4.58 | 103.6±11.11 | 9.4±3.72  |
>
> Here we provide the episodes (average of five trials with standard deviation) used to solve different environments ("solve" means the cumulative rewards exceed a certain threshold, which is 200 for Cartpole, -380 for Pendulum, -50 for Pusher, and -40 for Reacher). The environments with (r) indicate that the oracle reward function is provided.
>
>
> Some comments for this table:
>
> We use Bayesian linear regression (BLR) on the feature space in modeling (which is Gaussian Process with linear kernels). We want to clarify that **using the mean of BLR also provides some extent of exploration**, because the **covariance matrix**, which provides epistemic uncertainty in posterior sampling, is also used if we use mean in prediction. The formula for mean corresponds to a ridge regression instead of simple linear regression. The above can be verified in the posterior distribution formula of $w$ in our Section 5.1: Bayesian update and posterior sampling.
>
> However, **using mean does not explore as much as posterior sampling does**, since the posterior sampling adds randomness, besides the information provided by mean.
>
> Besides, there is currently no theoretical guarantee for Bayesian regret using the mean value instead of sampling: the posterior sampling lemma won't work if the agent simply uses mean, and it is hard to derive bounds directly on $\Delta_k$ in the paper (we only need to analyze $\tilde{\Delta}_k$ if we can use the posterior sampling algorithm).)
>
> Typos: we have fixed the typos and add some definitions. Thanks for pointing out!
>
> Please let us know if you have further questions about our response. We are very happy to have further discussions!

---

### Official Review · AnonReviewer4 · 2020-10-28
**Unclear theoretical novelty and missing baselines in the experiments.**

**Rating:** 5
**Confidence:** 4

**Review:**

Pros
--------
* The paper proposes a method to balance exploration and exploitation in reinforcement learning problems whose transitions and rewards are assumed to be sampled from Gaussian processes, and provide a Bayesian regret bound. Also, the paper shows how the proposed approach can be implemented in practice using model predictive control.

Cons
--------
* It is not clear what is the novelty of the theoretical results in this paper when compared to the regret bounds by Chowdhury & Gopalan (2019), who provide both frequentist and Bayesian bounds when the transitions and rewards are in an RKHS or sampled according to a GP. The bound of Chowdhury & Gopalan (2019) seem to be polynomial in the horizon H (their paper mention a $HSA\sqrt{T}$ bound in the particular case of finite MDPs), whereas the current paper says that "H is still unbounded" in their result. Hence, further clarification is needed regarding this point.
* The method is claimed to be computationally tractable: “it can be easily implemented by only optimizing a single sampled MDP”. However, the sampled MDP is continuous, and solving a continuous MDP is hard in general. Experimentally, the paper proposes the cross entropy method (CEM) for planning: in this case, planning is not exact and the regret bound does not hold anymore. I believe this issue should be made clear in the introduction/related work section.
* Although my main concern is the theoretical novelty, the experimental section can be improved: it would be interesting to compare the proposed approach to other strategies for exploration in deep RL, for instance
    * Bellemare et al. (2016), Unifying Count-Based Exploration and Intrinsic Motivation
    * Tang et al. (2017), #Exploration: A Study of Count-Based Exploration for Deep Reinforcement Learning
    * Azizzadenesheli et al. (2018), Efficient Exploration through Bayesian Deep Q-Networks;
and also perform experiments in small simple environments that satisfy the assumptions to check if the regret is sublinear (e.g. a continuous “grid world” with noisy transitions).
* In the nonlinear case (Section 4.2) it is not clear how to follow the steps of Yang & Wang (2019) to derive a Bayesian regret bound with feature representation, since their assumption is related to low-rank MDPs instead of Gaussian processes. In addition, it would be interesting to discuss how the model would be sampled (in Algorithm 1) in this case.


Suggestions & remarks
------------------------------------------------------
* Introduction: mention which of the cited papers proves the $H \sqrt{SAT}$ upper bound on the Bayesian regret, clarify whether $T$ is the number of episodes, or $H$ times the number of episodes.
* Some definitions are missing:
    * The linear kernel should be defined before Theorem 1
    * $M^k$ is not defined before appearing in Eq. 5
* Increase the font size of the text in Figure 1.
* In Algorithm 1, include what are the input parameters (e.g. $\sigma_r, \sigma_f$).
* Some suggestions for the proof:
     * Write the relation between $\Delta_k$ and $\tilde{\Delta}_k$.
    * Include (possibly in the appendix) more details about the arguments in the paragraph below Eq. 9. For instance, there is an argument about a bound on the information gain, which is not defined in the paper.  Also, it might be useful (for the reader) to restate (in the appendix) the results by Williams & Vivarelli (2000), Srinivas et al. (2012) and Russo & Van Roy (2014) required for the proof.

Typos
---------
* Abstract: $T$ instead of $\sqrt{T}$ in the regret bound with feature representation
* Page 9: Definition of MBPO, it should be “Model-Based Policy...”
* Typo in integration limits in Eq. 12   (the $\mu'-\mu$ at the bottom should be $\mu-\mu'$).

---

> ### Author Response · Authors · 2020-11-18
> **Rebuttal to R4: Theoretical novelty, computational tractability, clarification on baselines and linear assumptions with feature representation (Part 1)**
>
> Thanks for your review, suggestions, and detailed comments!  We respectfully disagree that our theoretical novelty is unclear (see Re1). And the model-free baselines you proposed are already outperformed by existing model-based algorithms for continuous state-action spaces (see Re3).
>
> **Re 1:  Theoretical novelty compared to the regret bounds by Chowdhury & Gopalan (2019), and other papers**
>
> We first clarify that the regret bound in Chowdhury&Gopalan(2019) can be exponential in $H$ in continuous spaces, and the $\tilde{O}(HSA\sqrt{T})$ bound in Chowdhury&Gopalan(2019) is for tabular MDPs. In continuous spaces, S and A are infinite, and we focus on **continuous state-action spaces** in our paper. Please check the general comments here for more details: **https://openreview.net/forum?id=asLT0W1w7Li&noteId=rJqD4DMG8jK**, where we show **why the regret bound in Chowdhury&Gopalan(2019) can be exponential in $H$ in continuous spaces in part 1.1, and clarify our improvement with respect to $H$ in part 1.2**. We also illustrate our improvement w.r.t. the dimension $d$ in part 2 and highlight other differences between our proof and Chowdhury&Gopalan(2019) in part 3.
>
> Note that **The best-known frequentist regret bound for continuous state-action space is $\tilde{O}(H^{3/2}d\sqrt{T})$** in Zanette et al.(2020) Theorem 1. Our Bayesian regret is $\tilde{O}(H^{3/2}d\sqrt{T})$, which is the best-known Bayesian regret for posterior sampling algorithms in continuous state-action spaces, and it also matches the best-known frequentist regret (Zanette et al.(2020) in our Related Work Section 2).
>
> **Re 2: Clarification on computational tractability**
>
> By computationally tractable, we are mainly comparing to Zanette et al.(2020), which achieve the same frequentist bound as ours. However, they use the UCB (upper confidence bound) algorithm and need to optimize over an upper confidence set of continuous MDPs, which is computationally prohibitive.
>
> Although solving continuous MDPs is generally hard, Chua et al. (2018) already show that MPC with CEM optimizer can solve Gym and Mujoco tasks within a handful of trials. Thus we conduct experiments in the **environments that MPC planning itself is shown sufficient to solve the MDPs** as in Chua et al.(2018). Using the same planning method, our algorithm outperforms theirs, which indicates the effectiveness of efficient exploration. Thank you very much for pointing out that the planning method is an approximate solution, and we have added this clarification in the introduction.

---

> > ### Author Response · Authors · 2020-11-18
> > **Part 2**
> >
> > **Re 3: Comparing with other exploration strategies already outperformed by the methods we compare with is unnecessary, and we have already shown sublinear regret in our results**
> >
> > Thanks for providing examples of other exploration strategies! These three papers mentioned here are all **model-free**, and the current state-of-the-art **model-based methods** compared in our paper (Chua et al.(2018), Janner et al. (2019)) have already **greatly outperform state-of-the-art model-free methods** in sample efficiency, as shown in their papers. Generally, model-based methods enjoys a significant advantage in sample efficiency than model-free methods. In our experiment, we have shown that our model-based algorithm outperforms Chua et al.(2018) and Janner et al. (2019), so we can safely expect that our algorithm can also outperform other model-free methods. Besides, our main focus is to solve **continuous state-action spaces** both theoretically and empirically. The experiments in Bellemare et al. (2016) and Azizzadenesheli et al. (2018) use DQN, in which the action spaces are limited to discrete spaces.
> >
> > Also, thanks for pointing out experiments in a continuous "grid world" to show sublinear regret! We want to point out that **in our experiment, we have already shown the sublinear regret**: for example, in Cartpole, we can observe that the agent finally achieves the best possible cumulative rewards (i.e. 200), and maintains the same performance afterwards (note that the dotted line in the experimental results Figure 1 indicates convergence results). Thus the regret after this episode would be zero, which means the regret is sublinear.
> >
> > **Re 4: Clarification on linear assumptions with feature representation, and model sampling methods**
> >
> > The low-rank MDP assumption actually refers to **linearity assumption with respect to feature representation**, as described in Assumption 1, Yang&Wang(2019). We have used linear kernels in Section 4.1 (which is actually **Bayesian linear regression**(BLR)). And in Section 4.2, instead of using BLR on the original state and action spaces, we use BLR on the feature representation of the original state-action spaces. That's why we can use their linear assumption on the feature space in Yang&Wang(2019). In our experiments, we extract the last hidden layer of neural networks as feature representation and conduct BLR on this feature space.
> >
> > For model sampling, we sample the MDP by **sampling the weights from the posterior distribution** for reward and transition BLR models respectively, and the details are described in Section 5.1: Bayesian update and posterior sampling.
> >
> > Thank you for pointing out the typos and other suggestions/clarifications for making this paper better! Specifically, $T$ is the number of total steps, which is consistent with the definition of all previous papers (Osband&Van Roy(2014), Chowdhury&Gopalan(2019), Yang&Wang(2019), Zanette et al.(2020)). $M^k$ is already defined before Equation (1).GP with linear kernels is Bayesian linear regression as described in Srinivas et al. (2012). We have made other revisions that are suggested in our paper.
> > Please let us know if you still have questions about our response. We are very happy to have further discussions.

---

> > > ### Comment · AnonReviewer4 · 2020-11-20
> > > **Need more clarification about Lipschitz constants in related work**
> > >
> > > Thank you for your response and clarifications! It is indeed very important to provide a regret bound that avoids a term that might be exponential in the horizon H.
> > >
> > > To check that I understood correctly, this improvement comes from the fact that you avoid using Lipschitz constants in your Lemma 1, whereas in the Lemma 7 of  Chowdhury & Gopalan(2019) (https://arxiv.org/pdf/1805.08052.pdf) the Lipschitz constant appears, is that correct?
> > > Please let me know if I'm missing something, but according to your Lemma 1, the constants $L_M$ and $L_*$ in their Assumption A1 would be $H\sqrt{ \frac{2}{\pi \sigma_f^2} }$, wouldn't they?  So their constant might be linear in H, and not exponential, under the assumptions you make. Using the fact that the value function is bounded by H, can't we get rid of the $\lambda_1$ in their LQR regret bound?
> > >
> > > In your paper, you mention that you "can avoid handling the Lipschitz continuity of the underlying value function", but in Assumption A1 in Chowdhury & Gopalan(2019) (https://arxiv.org/pdf/1805.08052.pdf), it is not really the Lipschitz constant of the *value function* that appears, it is the Lipschitz constant of the average value function over next states as a function of the next-state distribution.
> > >
> > > By the way, couldn't the proof of your Lemma 1 be simplified/improved by using Pinsker's inequality and the KL divergence between two Gaussian vectors?
> > >
> > > Do you think your improvement (with respect to the Lipschitz constant) only works for the Bayesian regret or do you think it could be extended to a frequentist setting?
> > >
> > > A minor point: You define (before Eq. 1)  an "optimistic policy under M". Did you mean "optimal policy"? There is also a typo in the argmax.

---

> > > > ### Author Response · Authors · 2020-11-20
> > > > **Further clarification in related work and our improvements**
> > > >
> > > > Thanks for your prompt response, suggestions, and request for the clarification! We have made it more clear in our explanation about the Lipschitz constant in part 1, official comment (https://openreview.net/forum?id=asLT0W1w7Li&noteId=rJqD4DMG8jK) and also in our introduction, for your reference. And here are our detailed responses w.r.t your questions:
> > > >
> > > > **Re 1: This improvement comes from the fact that you avoid using Lipschitz constants in your Lemma 1, whereas in the Lemma 7 of Chowdhury & Gopalan(2019) the Lipschitz constant $L\*$ appears, is that correct?**
> > > >
> > > > **The answer is no**. First, we need to clarify that **we did not claim that the use of $L\*$ (the Lipschitz constant for the future value function) yields exponential order in $H$**. Our point is, Chowdhury & Gopalan(2019) **makes the existence of $L\*$ as an assumption (A1), without clarifying the dependency on $H$ in $L\*$**: they said $L\*$ basically measures the connectedness of the MDP, and they claim that "all the results are stated assuming that the episode duration $H = O(lnT)$ for clarity, the explicit dependence on $H$ can be found in the theorem", but they totally ignore the dependency of $H$ in $L\*$, so in fact **they did not provide an explicit dependency on $H$ in their bound**. Also, instead of directly analyzing the upper bound of $L\*$ in the future value function, they use the Lipschitz constant of the **underlying value function** to bound $L\*$ in the example of LQR, which yields an exponential dependency of $H$ as we discussed. The main drawbacks of their paper are 1. there is no clear dependency of $H$, and 2. they derive a bound which is actually exponential in $H$ in the LQR example they provided, which is very loose. And our improvement in $H$ comes from Lemma 1 to clarify this dependency, and we derive a polynomial bound in $H$ with Lemma 1 and the proof from Lemma 1 to theorem 1, which matches the best-known frequentist regret bound.
> > > >
> > > > **Re 2: Their constant might be linear in H under the assumptions you make...**
> > > >
> > > > With the help of our Lemma 1, we can improve the result in Chowdhury & Gopalan(2019) as you mentioned, but we do not need extra assumptions. Except for this improvement, there are other theoretical improvements of our paper compared with Chowdhury&Gopalan(2019): please check part 2: lower dimensionality and part 3: fewer assumptions in this official comment https://openreview.net/forum?id=asLT0W1w7Li&noteId=rJqD4DMG8jK.
> > > >
> > > >
> > > > **Re 3: In your paper, you mention that you "can avoid handling the Lipschitz continuity..."**
> > > >
> > > > We did not misunderstand the future value function (the average value function over the next states as a function of the next-state distribution, as you mentioned) in A1 as we discussed in Re 1. By "without assuming the Lipschitz property of the underlying value function", we mean that both Osband&Van Roy(2014) and Chowdhury & Gopalan(2019) **bound $L*$ using the Lipschitz constant of the underlying value function in the example of LQR, not to mention they did not bound $L*$ in the main theorem**. And we clarify **the dependency on $H$ is exponential if we use the underlying value function to bound $L*$**, and we show that **this dependency can be significantly improved to polynomial with the subgaussian noise assumption** (which is already assumed in their papers, but not explored).
> > > >
> > > > **Re 4: Alternative proof of Lemma 1**
> > > >
> > > > Thank you for providing this idea. If we use Pinsker's inequality and the KL divergence between two Gaussians with a shared variance, the upper bound is independent of the variance $\sigma^2$. This result might serve as a tighter upper bound when $\sigma^2$ is relatively small, but when $\sigma^2$ is big, the upper bound in our proof would be tighter, so it cannot replace our proof. We will consider adding it as an alternative upper bound when $\sigma^2$ is small, but it does not affect the main theorem.
> > > >
> > > > **Re 5: Extending our improvement to frequentist regret**
> > > >
> > > > Our improvement can be extended to frequentist regret if we plug in our Lemma 1 as an upper bound of $L*$ in the analysis of Chowdhury&Gopalan(2019) in the frequentist regret bound for GP-UCRL. Also, even plugging in our Lemma 1 to their paper, our analysis would still differs from theirs: we directly analyze the regret bound of posterior sampling (PSRL) using the fact that the sampled and the real unknown MDP share the same distribution conditioned on history, while Chowdhury&Gopalan(2019) first analyze UCRL (Upper Confidence Bound in RL) and then transfer the result to PSRL. As a result, they need an extra assumption A2 (Regularity assumption) compared to us.
> > > >
> > > > Typos: Yes, thanks for pointing it out! It should be "optimal policy". We have fixed these typos.
> > > >
> > > > Please let us know if you still have questions about our response. We are very happy to have further discussions.

---

### Official Review · AnonReviewer3 · 2020-10-29
**Restricted contribution with no fundamental limit**

**Rating:** 5
**Confidence:** 4

**Review:**

The paper proposes a model-based posterior sampling algorithms with regret guarantees when the model is assumed to be drawn from a distribution randomly. The authors also provide numerical evaluations of the proposed method.

- The contribution of this work as theoretical work is limited. There is no study on fundamental limit. In addition, the performance guarantee seems worse than existing ones, although a fair comparison might be unavailable due to different technical assumes. However, the authors do not provide numerical comparison to existing algorithms with performance guarantee.

- It is not possible to assess the contribution from numerical comparison. There is no description on the hyperparameter selection of other algorithms (PETS, MDPO, SAC, ...). Hence, it is not reproducible as well.

- The definition of $BayesRegret$ seems incorrect as it takes $M^*$ as input argument. The authors need to describe what they mean by the expectation in eq. (3). In my understanding, $BayesRegret$ should take hyper-parameter to generate $M^*$ as input, while $Regret$ taking a random incidence of $M^*$ as input.

- I don't understand the meaning of regret bound $\tilde{O} (H^{3/2} d_\phi T)$ for non-linear case as regret of any algorithm is upper-bounded by $R_{max} T$.

---

> ### Author Response · Authors · 2020-11-17
> **Rebuttal to R3: Theoretical contribution & experimental details**
>
> Thanks for your review and detailed comments! We respectfully disagree that our theoretical guarantee is worse than existing ones with no fundamental limit. We have responded to each point as below. Please let us know if you have questions or suggestions.
>
> **Re 1: Performance guarantee compared with previous works**
> By "worse than existing ones", are you referring to regret bounds in tabular RL, which is $O(H\sqrt{SAT})$? (The study of previous works on tabular spaces in both frequenist and Bayesian settings are also mentioned in the introduction and related work.)
>
> Note that our analysis focuses on **continuous state-action spaces**, where $S$ and $A$ can go to infinity.
> The study of previous results on frequentist regrets is described in our Related Work (Section 2), where we show that the best result so far is $\tilde{O}(H^{3/2}d\sqrt{T})$. The study of previous works on Bayesian regrets is mentioned in our Introduction (Section 1), where we show that previous works fail to provide an exact dependency on episode length $H$.  A detailed explanation for theoretical improvements can be found in our general comments on the theoretical novelty above (**https://openreview.net/forum?id=asLT0W1w7Li&noteId=rJqD4DMG8jK**).
>
> Our Bayesian regret is $\tilde{O}(H^{3/2}d\sqrt{T})$, which is the best-known Bayesian regret for posterior sampling algorithms in continuous state-action spaces. our result also matches the best-known frequentist regret in continuous spaces (Zanette et al.(2020), described in related work),  where they have also proved that it matches the lower bound for this setting. However, their algorithm is based on optimization over an upper confidence set, which is computationally prohibitive, while posterior sampling algorithm only require optimizing a single MDP.
>
>
> **Re 2: Hyperparameters and other experimental details for reproducibility**
> Thanks for your suggestion! We have added experimental details for PETS, MBPO, SAC, and DDPG in Appendix 2. Hyperparameters for our experiments are also added for reproducibility.
>
> **Re 3: Definition for BayesRegret**
> Thank you for pointing it out! It is a typo: in $BayesRegret$ it should be $\phi$ instead of $M^*$, where $\phi$ is the prior distribution of $M^*$. We follow the same definition of $BayesRegret$ and $Regret$ in Osband&Van Roy (2017) Section 2. Here the expectation is taken with respect to the prior distribution of $M$. We have made the revision in the latest version.
>
> **Re 4: Regret bound typo in the abstract**
> We sincerely apologize for the typo in the abstract; it should be $\sqrt{T}$ as described in Section 4.2. We have made the revision in the latest version.
>
> Please let us know if you have additional questions about our response. We are very happy to have further discussions.

---

> > ### Comment · AnonReviewer3 · 2020-11-25
> > **Still not sure on the contribution**
> >
> > The revision clarifies a part of my concern from the critical typos. I increased my score from 4 to 5.
> >
> > However, I'm not sure on the theoretical contribution. In particular, BayesRegret is an upper bound of frequentist's worst-case regret. Hence, the authors' claim on the tightness of the proposed upper bound is not acceptable. Indeed, Osband&Van Roy(2017) translates the benefit from Bayesian setting in terms of the regret upper bound. In this sense, I'm not sure if the proposed algorithm is fully exploiting the benefit from considering MDP as a random variable. In addition, the analysis is heavily relying on Osband&Van Roy's one.

---

### Official Review · AnonReviewer1 · 2020-11-04
**Interesting idea, incremental contribution**

**Rating:** 5
**Confidence:** 4

**Review:**

This paper studies the model-based reinforcement learning. They propose a posterior sampling algorithm and provide a Bayesian regret guarantee. Under the assumption that the reward and transition functions are Gaussian processes with linear kernels, the regret bound is in the order of H^1.5 d sqrt{T} where H is the episode length. The dependence that regret is in polynomial of H is a nice property. However, this advantage seems to be obtained by the assumption of Gaussian processes with linear kernels. Besides, I have the following comments.

1) It seems that the result in Theorem 1 is quite straightforward from the results in Osband & Van Roy 2014. Could you justify your contribution in this result.

2) What is MPC? Suppose to be model predictive control? It is not officially introduced in the context.

3) In terms of computational complexity, could you elaborate more on the computational complexity of each line (component) of the algorithm. For general cases, posterior sampling could also be expensive as there are no closed-form solutions, one may need to use MCMC method.

---

> ### Author Response · Authors · 2020-11-17
> **Rebuttal to R1: Theoretical novelty and computational complexity**
>
> Thanks for your review and detailed comments! We respectfully disagree that our theoretical contribution is incremental.
>
> **Re 1: Our contribution in Theorem 1 compared to Osband&Van Roy(2014)**
> Our bound enjoys lower order in terms of episode length $H$ and dimensionality $d$ compared to Osband&Van Roy(2014). The improvement of $H$ mainly comes from the use of our Lemma 1 (which only make use of the noise assumption), instead of Gaussian processes with linear kernels. For a detailed analysis on our improvement of $H$ and $d$, please refer to our general comments on theoretical novelty above (https://openreview.net/forum?id=asLT0W1w7Li&noteId=rJqD4DMG8jK). Our result is NOT straightforward from results in previous works, since we provided an original Lemma 1 (proved in appendix), to significant improve the order of $H$ with the same assumption from the previous work, and the proof to use Lemma 1 to get Theorem 1 is also different from Osband&Van Roy(2014). Our bound is the best-known Bayesian regret for posterior sampling algorithms in continuous state-action spaces, and it also matches the best-known frequentist regret (Zanette et al.(2020)) in continuous spaces as mentioned in our Related Work (Section 2).
>
> **Re 2: What is MPC**
> MPC is model predictive control. Sorry for not citing the book in Section 5.2 (it should be Camacho&Alba (2013): "Model predictive control". We have made a revision for this. Thank you for pointing out!
>
> **Re 3: Computational complexity on each component of the algorithm**
> The main cost of computation comes from MPC planning and posterior update of the Bayesian model.
>
> We use MPC action selection with a CEM optimizer as in Chua et al.(2018), to solve a single sampled MDP. At each step in an episode, we need to calculate the predicted return from the current step up to $\tau$ steps(which is the planning horizon, usually $\tau=30$). Using CEM, we generate $M$*$\tau$ actions to form $M$ action sequences at each iteration, where $M$ is the population size (usually 500), use the sampled MDP to generate random returns, and select several elite sequences to update the CEM distribution to use in next iteration (the max iteration number is usually 5). So in each episode, the complexity of computation from MPC is $O(\tau MH)$. Overall, it enjoys a lower complexity than Chua et al.(2018) since we do not use an ensemble of different models.
>
> Posterior update: The matrix multiplication to calculate of matrix $A$ in 5.1 is $O(d^2N)$, and the inverse of matrix $A$ is $O(d^3)$, where $N$ is the number of data points, and $d$ is the dimension of the last hidden layer of the neural network. In our experiments, we find forcing the last hidden layer dimension to be the same as the input dimension works well in our modeling, so the posterior update is not computationally expensive.
>
> Please let us know if you have additional questions. We are very happy to have further discussions.

---

### Author Response · Authors · 2020-11-17
**(This part has been added to our introduction to highlight our theoretical novelty) We achieve the first Bayesian regret bound for posterior sampling algorithms that is simultaneously near optimal in T and polynomial in episode length H in continuous state-action spaces**

Most reviews question our novelty compared to previous bounds of continuous PSRL in Osband&Van Roy(2014) and Chowdhury&Gopalan(2019). We have the following key points that support our claim:

## 1. Previous bounds of **continuous PSRL**  can be **exponential in H**, while in our analysis **we significantly improved it to polynomial**:

### 1.1 The dependency on $H$ by the Lipschitz constant in previous works:

In Osband&Van Roy(2014) , the bound is  $\tilde{O}(\sigma_R\sqrt{d_K(R)d_E(R)T}+\mathbb{E}[L^*]\sigma_p\sqrt{d_K(P)d_E(P)})$, where $L^*$ is a global Lipschitz constant for the future value function defined in their eq. (3). However, $L^*$ is dependent on H: the difference between the initial states will propagate in $H$ steps, which results in a term dependent of $H$ in the value function. The authors do not mention this dependency, so **there is no clear dependency on $H$ in their regret**. Moreover, they use the Lipschitz constant of the **underlying value function as an upper bound** of $L^*$ in the corollaries , which yields an **exponential order** in $H$: take their Corollary 2 of linear quadratic systems as an example: the regret bound is $\tilde{O}(\sigma C\lambda_1n^2\sqrt{T})$, where $\lambda_1$ is the largest eigenvalue of the matrix $Q$ in the optimal value function $V_1(s) = s^TQs$, where $V_1$ denotes the value function counting from step 1 to H within an episode, $s_{i+1} = As_i+Ba_i + \epsilon_{P,i},$ and $r_{i} = s_i^TPs_i+a_i^TRa_i + \epsilon_{P,i}$, $i\in [H]$. However, the eigenvalue of $Q$ is actually exponential in $H$: recall the Bellman equation we have $V_i(s_i) = \min_{a_i} s_i^TPs_i + a_i^TRa_i + \epsilon_{P,i} +V_{i+1}(As_i+Ba_i+\epsilon_{P,i})$, $V_{H+1}(s)=0$ . Thus in $V_1(s)$, there is a term of $(A^{H-1}s)^TP(A^{H-1}s)$, and the eigenvalue of the matrix $(A^{H-1})^TPA^{H-1}$ is exponential in $H$. **Even if we change the reward function from quadratic to linear**, say $r_{i} = s_i^TP+a_i^TR + \epsilon_{P,i}$, we still get a term of $(A^{H-1}s)^TP$ in $V_1(s)$, which means the Lipschitz constant of the underlying value function is **exponential in H**.

In  Chowdhury&Gopalan(2019), they maintain the same assumption of Lipschitz property as in Osband&Van Roy(2014),  therefore we can see $\mathbb{E}[L^*]$ in the bound without explaining its dependency on $H$. As a result, there is still no clear dependency on $H$ in their regret, and in their Corollary 2 of LQR, they follow the same steps as Osband&Van Roy(2014), and still maintain the term of $\lambda_1$, which is actually **exponential in H** as discussed.

### 1.2 Our improvement:

In our analysis, we only use the property of subgaussian noise, which is already assumed in Osband&Van Roy(2014) and  Chowdhury&Gopalan(2019), **to develop a clear polynomial order in H, without assuming the Lipschitz continuity of the underlying value function**. More specifically, we prove Lemma 1, and use it to develop a clear dependency on H, thus we can avoid handling the Lipschitz continuity of the underlying value function.

## 2. **Lower dimensionality**:

We use feature embedding to lower the dimensionality in previous Bayesian regret bounds. The eluder dimension of neural networks in Osband&Van Roy(2014)  can blow up to **infinity**, and the information gain used in Chowdhury&Gopalan(2019) yields **exponential order of dimension $d$ if nonlinear kernels are used**, but linear kernel can only model linear functions, thus the representation power is highly restricted if the polynomial order of d is desired.

In our analysis, we first derive the result for **linear kernels**, and increase representation power of our model by extracting the last hidden layer of neural networks as features and construct a Bayesian linear regression on the feature space, and thus we can use the result of linear kernels to derive a bound in nonlinear cases, and obtain a **linear order of the feature dimension** in the bound. The feature dimension, i.e. the dimension of the last hidden layers required for modeling is much lower than the exponential of the input dimension, so our bound enjoys lower dimensionality.

## 3. **Fewer assumptions and different proof strategies compared to Chowdhury&Gopalan(2019):**

Although we also use kernelized MDP like Chowdhury&Gopalan(2019), we **omit their assumptions A1 (Lipschitz assumption) and A2 (Regularity assumption)**. A1 is not necessary due to our Lemma 1 and we improve the regret bound in H as discussed in part 1. A2 is also omitted: we directly analyze the regret bound of PSRL **using the fact that the sampled and the real unknown MDP share the same distribution conditioned on history**. However, Chowdhury&Gopalan(2019) first analyze UCRL (Upper confidence bound in RL) and then **transfer the result of UCRL to PSRL**. So they need an extra assumption A2 compared to ours. The only overlap between our analysis and theirs is the use of information gain to use the concentration property of GPs.

---

### Author Response · Authors · 2020-11-23
**Summary of revisions**

We are grateful for all reviewers' comments and here is a summary of revisions we have made accordingly to highlight our contribution compared with previous works, further facilitate reproducibility and improve the overall readability of the paper:

1. We add explanations on why previous works with Bayesian regrets are limited in the order of $H$ and $d$ (as summarized in the comments here  https://openreview.net/forum?id=asLT0W1w7Li&noteId=rJqD4DMG8jK), and provide bullet points to highlight our **theoretical contributions** compared to these works in the introduction section.

2. We add hyperparameters and other experimental details for compared baselines and our approach in Appendix 2 to facilitate reproducibility.

3. We fix all the typos pointed out by reviewers:

Reviewer 1: fix the reference for MPC;

Reviewer 2: replace \sigma_R with \sigma_r /, \sigma_P with \sigma_f; fix d in theorem 1; definition of \delta_k(r) and \delta_k(f);

Reviewer 3: fix the typo in the order of $T$ in the abstract for the regret bound with feature representation; fix the input parameter for BayeRegret;

Reviewer 4: add an explanation for linear kernels; add clarification of $T$ and fix the typo in the order of $T$ in the abstract for the regret bound with feature representation; fix integration limits in Appendix 1.

---

### Decision · Program_Chairs · 2021-01-07
**Final Decision**

**Decision:**

Reject

**Comment:**

This paper presents a model-based posterior sampling algorithm in continuous state-action spaces theoretically and empirically. The work is interesting and the authors provide numerical evaluations of the proposed method. But the reviewers find the contribution of the work limited.